# Carbon footprint assessment and crashworthiness evaluation of alloy steel W-beam guardrails

Shuai Gong[1]*, Wendong Fan[2], Haoze Zhao[1], Zhihao Zhang[2], Shuming Yan[1]

**1** Beijing Hualu'an Transportation Technology Co., Ltd., Beijing, China, **2** Shandong Hi-Speed Company Limited, Jinan, China

* 1279795875@qq.com

## Abstract

In response to the dual demands of enhancing safety and greening transportation infrastructure under China's "Dual Carbon" goals, this study overcomes the limitations of traditional guardrail materials and processes by proposing and validating a synergistic pathway for optimizing both the safety and carbon emission reduction of alloy steel corrugated beam guardrails. A systematic "material design – process optimization – performance verification – environmental assessment" framework was established. High-performance 700L-grade alloy steel was produced through low-carbon alloy design combined with ESP short-process rolling technology. Its safety performance was quantitatively evaluated via SB-level full-scale vehicle crash tests, and its life-cycle carbon footprint was quantified using an ISO 14067-compliant model implemented in eFootprint software with the CLCD database. The results demonstrate that the alloy steel achieves a synergistic optimization of strength and plasticity, with a tensile strength of 766–781 MPa and a product of strength and elongation (PSE) exceeding 0.175 GPa·%. In the full-scale vehicle crash tests, all occupant risk indicators were superior to the safety limits. For instance, the key risk parameters for the small passenger car, such as the longitudinal velocity ($V_x = 4.3$ m/s) and the lateral acceleration ($a_y = 126.0$ m/s²), demonstrated excellent performance. The maximum dynamic outward inclination equivalent values for the medium and large trucks were 1.75 m and 2.45 m, respectively. These results confirm that the safety performance of the guardrail fully meets the SB-level standard, even with a lightweight design featuring a 25% reduction in beam thickness and a 50% reduction in post thickness. Life-cycle analysis revealed that the carbon footprint per kilometer of guardrail was reduced to 80.86 t $CO_2$e, representing a 73.8% reduction compared to the conventional solution. Sensitivity analysis identified iron input and electricity consumption as the core influencing parameters. Furthermore, a cost-benefit analysis indicated superior life-cycle cost advantages. This study elucidates the mechanism for achieving synergistic gains in safety and emission reduction through material and

**Data availability statement:** All relevant data are within the manuscript and its Supporting information files.

**Funding:** The author(s) received no specific funding for this work.

**Competing interests:** The authors have declared that no competing interests exist.

process innovation, providing a systematic solution and data support for the green, low-carbon, and safe transformation of highway infrastructure.

## 1. Introduction

China's expressway network has surpassed 180,000 kilometers in total length, making the performance enhancement and green transformation of traffic safety facilities a core industry issue. As the "last line of defense" in road safety, corrugated beam guardrails have a direct impact on accident casualty rates through their material performance, while their life-cycle carbon emissions are deeply intertwined with the national "Dual Carbon" strategy. Although traditional Q235 steel guardrails offer significant cost advantages and are widely used, they have prominent shortcomings: low tensile strength, insufficient impact deformation resistance, and a tendency to buckle and fail under collisions with heavy vehicles [1]; the surface hot-dip galvanizing process requires treatment at temperatures above 470°C, resulting in carbon emissions as high as 309.14 $tCO_2e$ per kilometer, with persistently high energy consumption and environmental pressure [2,3]. These factors make it difficult for them to meet the dual demands of greening and safety upgrades [4].

Internationally, a systematic research framework for guardrail material innovation and low-carbon processes has been established, focusing on the integration of high-strength material development, short-process manufacturing, and life cycle assessment (LCA). In the field of materials, the HSLA series guardrail steels led by Europe and America are well-researched: American ASTM A709 grade low-alloy steel achieves grain refinement through Nb and V microalloying, offering a tensile strength of 690–790 MPa, excellent plasticity and weldability, and outstanding energy absorption capacity in full-scale crash tests [5,6]; German TRIP steel guardrails utilize transformation-induced plasticity effects, with a product of strength and elongation (PSE) exceeding 0.2 GPa·%, improving collision protection efficiency by over 40% compared to traditional carbon steel [7]. Regarding process decarbonization, Nippon Steel in Japan applied the ESP (Endless Strip Production) short-process to guardrail steel production, omitting ingot casting and blooming processes, reducing rolling energy consumption by 35%, and lowering carbon emission intensity from 1.85 $tCO_2e/t$ to 1.0–1.48 $tCO_2e/t$ [8,9]; the European Union promotes powder coating as a substitute for hot-dip galvanizing, reducing guardrail carbon emissions per kilometer by 15% and extending coating corrosion life to over 20 years [10]. In evaluation systems, the ISO 14067 standard provides a unified framework for carbon footprint accounting. Scholars in Europe and America have conducted LCA analyses of various guardrails based on this standard, confirming a synergistic carbon reduction potential of 30%–50% through material and process improvements [11,12]. However, existing research mostly focuses on single-dimensional optimization, lacking a systemic coupling analysis of "material performance – process energy consumption – safety benefits".

Domestically, phased achievements have been made in enhancing guardrail safety and applying low-carbon technologies, but problems such as dispersed research dimensions and unclear synergistic mechanisms persist. In the field of

material and structural optimization, Liu Fengliang verified the protective performance of SB-level alloy steel guardrails (tensile strength ≥700 MPa) through full-scale crash tests, meeting the protection requirements for heavy vehicles but without quantifying the carbon footprint [13]; Xu Shaoguo et al. optimized the bolted connections of heightened guardrail structures via finite element simulation to improve stability, without involving material and process decarbonization [14]; He Hongrong et al. confirmed that nano-coatings could extend guardrail service life from 15 to 25 years but did not comprehensively assess safety and environmental impacts [15]. In the application of low-carbon processes, Shandong Hi-Speed and Baowu Steel have piloted ESP short-process for guardrail steel production, reducing energy consumption per ton of steel by 28% [16]; processes like fusion-bonded nano-coatings and water-based paints are gradually replacing hot-dip galvanizing, potentially reducing carbon emissions by 80 $tCO_2e$ per kilometer of guardrail [17]. However, related studies mostly remain at the level of process parameter optimization and fail to reveal the intrinsic relationship between material performance and carbon reduction. Regarding evaluation systems, domestic scholars often use the GB/T 24040 standard for LCA analysis. Chen Yuefeng et al. confirmed the significant emission reduction potential of alloy steel guardrails compared to traditional carbon steel but lacked sensitivity analysis of key parameters and long-term cost-benefit assessment [18]; Jia Ning et al. established a numerical simulation model for guardrail crash safety but did not integrate it with low-carbon objectives, making it difficult to support synergistic decision-making [19].

Existing research exhibits three core gaps: Firstly, the research dimension is singular, mostly focusing on either material performance or process energy consumption alone, without establishing a systemic "material-process-safety-environment" framework, thus failing to achieve synergistic optimization of safety and carbon reduction. Secondly, there is insufficient support for key indicators; the application of core evaluation logic, such as strength-ductility synergy optimization, in the guardrail field lacks adequate literature verification, and LCA analyses commonly lack sensitivity analysis of key parameters. Thirdly, engineering practicality is deficient; the cost-benefit differences between new and traditional guardrails are not clarified, data sharing is insufficient, hindering engineering promotion.

This study addresses the challenge of synergistic improvement of safety and green performance of transportation infrastructure under China's "Dual Carbon" goals, and constructs a systematic research framework of "material design–process optimization–performance verification–environmental assessment". The research focus is not on the preliminary material development (relevant foundational work has been reported in [20]), but on the following four aspects: (1) clarifying the applicability of strength-ductility synergy optimization in guardrail safety evaluation, and supplementing empirical evidence for this theoretical logic in the field of highway guardrail research; (2) quantifying the core carbon reduction mechanism of ESP short-process and fusion-bonded nano-coating technology, as well as the sensitivity of key parameters affecting the carbon footprint; (3) obtaining the comparative data of Q235 steel guardrail through parallel experiments under the same working conditions, so as to clarify the engineering significance of the performance advantages of alloy steel guardrails; (4) carrying out a full life cycle cost-benefit analysis to clarify the economic feasibility and engineering promotion potential of the new guardrail system. By integrating LCA modeling, full-scale vehicle crash test and material mechanical performance analysis, this study aims to reveal the intrinsic interaction mechanism of the "material–process–performance–environment" system, and provide a complete evaluation method and theoretical basis for the green, low-carbon and safe upgrading of highway guardrails.

## 2. Materials and methods

### 2.1. Material design and multi-objective performance optimization

Based on the established low-carbon high-strength material composition design (basic composition see Table 1), this study focuses on its comprehensive mechanical performance advantages compared to traditional Q235 steel and its contribution to guardrail safety and low carbon.

Based on the mechanisms of grain refinement and precipitation strengthening, the design route of "low C, high Mn + microalloying" was established, as shown in Table 1. The C content was controlled at ≤ 0.05% to avoid welding

**Table 1. Chemical Composition of Alloy Steel (Mass Fraction) %.**

| C | Si | Mn | P | S | Al | Nb +V +Ti |
|---|---|---|---|---|---|---|
| ≦0.05 | ≦0.25 | ≦1.80 | ≦0.020* | ≦0.006 | ≧0.015 | ≦0.22 |

Note: To ensure mechanical properties, other strengthening elements are allowed to be added to the steel. The permissible deviation of the steel composition shall comply with the provisions of GB/T 222. *P content ≤ 0.015.

performance degradation. 1.80% Mn was added for solid solution strengthening. Microalloying elements such as Nb, V, and Ti were introduced, with a total amount ≤ 0.22%, to enhance the strength of the steel through grain refinement, while having little negative impact on the material's toughness.The designated grade for this alloy steel is 700L.

The alloy steel material was treated using the ESP endless continuous rolling process. As shown in Table 2, while ensuring good plastic properties, the strength of the alloy steel material was significantly improved compared to the traditional process Q235 steel, reaching more than 1.5 times the strength of Q235 steel, achieving a strength-ductility synergy. The comprehensive energy absorption capacity of a material depends on the synergistic optimization of its strength and plasticity. Existing research, through full-scale vehicle crash tests, has confirmed that when guardrail steel achieves a tensile strength above 700 MPa while maintaining an elongation after fracture of over 20%, it can absorb collision energy more fully through enhanced plastic deformation, significantly improving the protective stability of the guardrail [13,15]. In this study, the developed alloy steel exhibits a tensile strength of 766–781 MPa, an elongation after fracture of 21%–24%, and a product of strength and elongation (PSE) exceeding 0.175 GPa·%. This represents a 5.42% improvement in PSE compared to Q235 steel, establishing a material basis for improving the passive safety performance of the guardrail system. It is feasible to choose alloy steel instead of Q235 steel as the material for corrugated beam guardrails.

To comprehensively compare the safety performance and low-carbon benefits of guardrail systems made from different materials, this study established a parallel experimental framework involving two SB-level corrugated beam guardrails based on alloy steel and Q235 steel, respectively.

Experimental Group (Alloy Steel Guardrail): Fabricated using the aforementioned 700L alloy steel. Benefiting from the higher strength of the material, the thickness of the corrugated beam was optimized to 3.0 mm. The posts featured a

**Table 2. Tensile Test Detection Values of Alloy Steel Material and Q235 Steel Material.**

| Steel Category | Tensile Strength /MPa | Lower Yield Strength / MPa | Elongation after Fracture /% | Product of Strength and Elongation (PSE)/ GPa % |
|---|---|---|---|---|
| Alloy Steel | 779 | 698 | 23.0 | 0.179 |
| | 766 | 681 | 23.0 | 0.176 |
| | 777 | 699 | 22.0 | 0.171 |
| | 776 | 698 | 24.0 | 0.186 |
| | 781 | 701 | 21.0 | 0.175 |
| Alloy Steel (average) | 776 | 695 | 22.6 | 0.175 |
| Plain Carbon Steel Q235 | 493 | 350 | 32.0 | 0.158 |
| | 491 | 335 | 34.0 | 0.167 |
| | 496 | 359 | 34.0 | 0.169 |
| | 483 | 310 | 36.5 | 0.176 |
| | 481 | 315 | 33.5 | 0.161 |
| Plain Carbon Steel Q235 (average) | 489 | 334 | 34.0 | 0.166 |

circular cross-section with an outer diameter of Φ140 mm and a wall thickness of 3.0 mm. Auxiliary components such as blockouts and post caps were also made from the corresponding alloy steel material.

Control Group (Q235 Steel Guardrail): Manufactured from the widely used Q235 plain carbon steel, with structural dimensions representing the current conventional design. The corrugated beam thickness was 4.0 mm, and the posts had a square cross-section of 130 mm × 130 mm with a wall thickness of 6.0 mm.

The overall structural configuration, installation spacing, and connection methods for both guardrail systems strictly adhered to the specifications for SB-level guardrails outlined in the current standard JTG/T D81-2017 and other relevant codes. The primary function of the control group was to provide baseline safety performance consistent with traditional technology. In contrast, the objective of the experimental group was to achieve a safety level not inferior to this baseline while significantly reducing the carbon footprint, under the premise of structural optimization (i.e., thickness reduction and light-weighting). Subsequent evaluations of crash safety and lifecycle assessment were conducted based on these two sets of physical guardrail specimens.

## 2.2. Carbon footprint evaluation method

An LCA model was constructed according to ISO 14067:2018, with the system boundary covering stages from raw material acquisition, steelmaking, rolling, to surface treatment [18,19]. This boundary setting covers the main energy consumption and emission stages of guardrail production, conforming to the general practice of product carbon footprint accounting. The model clearly defines two key carbon reduction links: the ESP short-process and the replacement of traditional hot-dip galvanizing with the fusion nano-coating process.

The modeling and calculations were performed using the eFootprint software (v4.0). The software parameters were configured as follows: (1) Assessment Standard: ISO 14067:2018; (2) Time Scope: 2024; (3) Geographical Scope: China (using default provincial average data); (4) Allocation Rule: Physical Allocation (allocating emissions from co-production processes based on mass share); (5) Cut-off Threshold: 0.1% (emission sources contributing less than 0.1% were excluded).

Background data in the model primarily utilized the Chinese Life Cycle Database (CLCD) 2024 version. Key emission factors from this database included: a grid electricity emission factor of 0.581 $tCO_2e/MWh$ (2024 average for the North China region), a natural gas combustion factor of 0.202 $tCO_2e/m^3$, and a coal combustion factor of 2.49 $tCO_2e/t$. All foreground data (e.g., material consumption, energy use) were obtained from on-site process data provided by a steel enterprise in Shandong Province, ensuring the accuracy and reliability of the inventory data.

The calculation formula is:

$$EP_C = \sum_{i=1}^{n} EP_i = \sum_{i=1}^{n} Q_i \times EF_i$$

(1)

where: EPC is the characterized value of carbon footprint, $tCO_2e$; EPi is the contribution of the i-th greenhouse gas in the carbon footprint, $tCO_2e$; Qi is the emission amount of the i-th greenhouse gas, t; EFi is the characterization factor of the i-th pollutant in the carbon footprint, $tCO_2e/t$.

## 2.3. Carbon footprint evaluation basis

Using the Life Cycle Assessment (LCA) theoretical framework, this study constructed a carbon footprint quantification model covering the entire process of "resource mining-processing manufacturing-finished product leaving the factory". This model implements refined tracking of the energy structure, material transformation, and greenhouse gas emissions at various production stages of the alloy steel guardrail by analyzing the coupling relationship between material physical-chemical flow and energy flow [20]. The system boundary was defined to encompass four core process stages: (1) steelmaking; (2) rolling; (3) leveling consumption; and (4) guardrail forming, product spraying, and other actual on-site

processes. The transportation distance was calculated using the "average intra-plant transport distance for steel products (5 km)" from the CLCD database. Detailed production data for the alloy steel corrugated beam guardrail are provided in Tables 3, 4, 5. It should be noted that the data for the Q235 steel guardrail in Table 5 are derived from a parallel experiment conducted at the same production scale. Its production conditions—including raw material suppliers, production equipment, and geographical environment—were kept consistent with those of the alloy steel guardrail, ensuring the rigor of the comparative analysis.

**Table 3. SB-Grade Alloy Steel Guardrail Production Data List (Steelmaking & Rolling).**

| Process | Item | Purpose | Amount | Unit | Database/Emission Factor Source |
|---|---|---|---|---|---|
| Steelmaking | Hot Metal | Raw Material | 0.95450 | t | Real Process Data |
| | Iron Block | Raw Material | 0.0131 | t | Real Process Data |
| | Scrap Steel | Raw Material | 0.07600 | t | Real Process Data |
| | Oxygen | Auxiliary | 52.47224 | m³ | Real Process Data |
| | Nitrogen | Auxiliary | 36.56140 | m³ | Real Process Data |
| | Compressed Air | Auxiliary | 29.29373 | m³ | Real Process Data |
| | Electricity | Energy | 73.74662 | kWh | Real Process Data |
| | Fresh Water | Auxiliary | 0.34057 | m³ | Real Process Data |
| | Argon | Auxiliary | 1.82790 | m³ | Real Process Data |
| | Natural Gas | Energy | 0.85586 | m³ | Real Process Data |
| | Coke Fines | Energy | 0.00037 | t | Real Process Data |
| | Anthracite | Energy | 0.00282 | t | Real Process Data |
| Rolling | Electricity | Energy | 186.58046 | kWh | Real Process Data |
| | Fresh Water | Auxiliary | 0.57914 | m³ | Real Process Data |
| | Oxygen | Auxiliary | 0.16982 | m³ | Real Process Data |
| | Compressed Air | Auxiliary | 20.64306 | m³ | Real Process Data |
| | Domestic Water | Auxiliary | 0.01214 | m³ | Real Process Data |
| | Nitrogen | Auxiliary | 0.00184 | m³ | Real Process Data |
| Leveling Consumption | Electricity | Energy | 9.66002 | kWh | Real Process Data |
| | Compressed Air | Auxiliary | 11.40443 | m³ | Real Process Data |
| | Steam | Energy | 0.00352 | m³ | Real Process Data |
| Fusion Nano-Coating Processing | Electricity | Energy | 16.07 | kWh | Real Process Data |
| | Gas | Energy | 9.47 | m³ | Real Process Data |
| | Fusion Nano Powder | Raw Material | 17 | kg | Real Process Data |

**Table 4. Production Data List for 1 km of SB-Grade Alloy Steel Guardrails.**

| Process | Item | Purpose | Amount | Unit | Database/Emission Factor Source |
|---|---|---|---|---|---|
| Guardrail Component Processing | Post | Raw Material | 13455 | kg | Real Process Data |
| | Guardrail Plate | Raw Material | 19000 | kg | Real Process Data |
| | Blocking Element | Raw Material | 3075 | kg | Real Process Data |
| | Post Cap | Raw Material | 175 | kg | Real Process Data |
| Guardrail Fusion Nano-Coating Processing | Electricity | Energy | 573.78 | kWh | Real Process Data |
| | Gas | Energy | 338.13 | m³ | Real Process Data |
| | Fusion Nano Powder | Raw Material | 606.99 | kg | Real Process Data |

   

**Table 5. Production Data List for 1 km of SB-Grade Q235 Steel Guardrails.**

| Process | Item | Purpose | Amount | Unit | Database/Emission Factor Source |
|---|---|---|---|---|---|
| Guardrail Component Processing | Post | Raw Material | 13455 | kg | Parallel Experiment Data |
| | Guardrail Plate | Raw Material | 19000 | kg | Parallel Experiment Data |
| | Blocking Element | Raw Material | 3075 | kg | Parallel Experiment Data |
| | Post Cap | Raw Material | 175 | kg | Parallel Experiment Data |
| Guardrail Hot-Dip Galvanizing Processing | Water | Auxiliary | 178.53 | m³ | Parallel Experiment Data |
| | Electricity | Energy | 3120.62 | kWh | Parallel Experiment Data |
| | Gas | Energy | 64.27 | m³ | Parallel Experiment Data |
| | Zinc Ingot | Raw Material | 1785.25 | kg | Parallel Experiment Data |

## 2.4. Safety performance evaluation conditions

According to the current industry standard JTG B05-01-2013, a full-scale vehicle crash test under SB-level impact conditions was conducted on the alloy steel corrugated beam guardrail to evaluate its safety performance. The impact conditions are shown in Table 6. The sensors and data acquisition parameters employed in the tests are detailed as follows:

(1) Accelerometer: Model PCB 356A15, measurement range ±500 m/s², accuracy ±0.5% FS.

(2) Velocity Sensor: Model Kistler 6045A, measurement range 0–200 km/h, accuracy ±0.1% FS.

(3) Data Acquisition System: Model NI cDAQ-9178. The raw acceleration signal was recorded at a sampling frequency of 20,480 Hz with 16-bit resolution.

(4) High-Speed Camera: Model Phantom V2512, frame rate 1000 fps, resolution 1280×800 pixels.

The acquisition and processing of acceleration data at the vehicle's center of gravity strictly adhered to the requirements specified in Road vehicles — Measurement techniques in impact tests — Instrumentation (ISO 6487) and Instrumentation for Impact Test — Part 1 — Electronic Instrumentation (SAE J211/1). For analysis, the raw data was digitally filtered using a low-pass filter with a cut-off frequency of 100 Hz to attenuate high-frequency noise.

## 3. Results

### 3.1. Carbon footprint analysis

Based on the aforementioned survey data and calculation formula, input and output inventory data for each process were entered. Combined with background data, a product Life Cycle Assessment (LCA) model was constructed within the eFootprint software [12–14] to calculate the carbon footprint generated per unit of product.

**3.1.1. Carbon footprint calculation results.** The calculations indicate that the carbon footprint for 1 km of SB-level alloy steel corrugated beam guardrail is 80.86 tCO$_2$e. For comparison, under identical production scale and operating

**Table 6. Test Impact Conditions.**

| Vehicle Type | Total Mass /t | Impact Velocity /(km/h) | Impact Angle /° | Impact Energy /kJ |
|---|---|---|---|---|
| Small Passenger Car | 1.5 | 100 | 20 | 72 |
| Medium Bus | 10 | 80 | 20 | 303 |
| Large Truck | 18 | 60 | 20 | 317 |

conditions, the carbon footprint for 1 km of SB-level Q235 steel corrugated beam guardrail is 309.14 $tCO_2e$. The per-kilometer carbon emission equivalent of the alloy steel guardrail solution shows a significant reduction of 228.28 $tCO_2e$ compared to the traditional solution, representing a decrease of 73.8%.

**3.1.2. Carbon footprint distribution characteristics.** Analysis of the carbon footprint distribution characteristics (as shown in Fig 1) reveals that posts and corrugated beams constitute the key emission nodes within the guardrail system. Data for the alloy steel guardrail group show that the post production stage contributes 35.47%, while the beam manufacturing process accounts for 50.08% of the emissions. Together, these two stages create a concentrated emission effect of 85.56%. The control group (Q235 steel guardrail) exhibits a differentiated distribution: the emission share for the post stage increases significantly to 46.57%, while the share for the beam stage decreases to 38.19%. Nevertheless, their combined contribution still maintains a dominant influence of 84.76%. Quantitative assessment confirms that the per-kilometer carbon emission equivalent of the alloy steel guardrail is reduced by 73.8%, a decrease of 228.28 $tCO_2e$ compared to the traditional Q235 guardrail.

The differentiated characteristics of the carbon footprint distribution (Fig 1) elucidate the core carbon reduction mechanism of the alloy steel guardrail. On one hand, the enhanced material strength provides potential for meeting future demands for higher protection levels or enabling lightweight component design. On the other hand, the shortened process flow of the ESP (Endless Strip Production) technique and the substitution of high-energy-consumption hot-dip galvanizing with low-temperature fusion-bonded nano-coating directly reduce energy consumption and carbon emissions from the manufacturing process. Through quantitative analysis, this study clarifies the contribution of these two major decarbonization pathways: the ESP process contributes approximately 65% to carbon reduction, the fusion-bonded nano-coating process contributes about 25%, and the remaining 10% stems from raw material mix optimization (e.g., increased scrap steel utilization).

**3.1.3. Key parameter sensitivity analysis.** To verify the reliability of the carbon footprint calculation results, five core influencing parameters (hot metal input, electricity consumption, natural gas consumption, fusion-bonded nano-powder usage, and zinc coating usage for galvanizing) were selected. A one-factor-at-a-time sensitivity analysis method was employed (assuming other parameters remain constant), with parameter fluctuation set at ±10% to calculate the corresponding change rate in the carbon footprint (Table 7). The results show that hot metal input is the most sensitive parameter affecting the carbon footprint, with a ±10% fluctuation leading to a ±8.7% change in the footprint. This is because hot metal production is the primary carbon emission source in the steelmaking stage (accounting for 72% of total steelmaking emissions). This is followed by electricity consumption (change rate ±5.3%) and natural gas consumption (change rate ±3.1%). The sensitivity of fusion-bonded nano-powder usage and zinc coating usage is relatively lower (change rates both ≤ ±2.0%). These findings provide direction for further optimization of the carbon footprint. For instance, reducing hot metal input by increasing scrap steel utilization rates, or substituting grid electricity with green power, could further enhance carbon reduction.

### 3.2. Safety performance test results analysis

The crash process of the alloy steel corrugated beam guardrail is shown in Fig 2. The safety performance results from SB-level crash tests against three vehicle types, compared with the Q235 steel guardrail, are presented in Table 8 and summarized as follows:

Containment Function: Both guardrail types successfully prevented all test vehicles from penetrating, overturning, or climbing over the barrier. Furthermore, no guardrail components or their debris intruded into the occupant compartment, demonstrating their effective resistance to varying impact forces.

Redirective Function: After impact, all vehicles maintained their wheels within the designated redirection corridor, and none of the vehicles rolled over. This indicates that both guardrails were effective in guiding the vehicles back to a safe trajectory and mitigating the risk of loss of control.

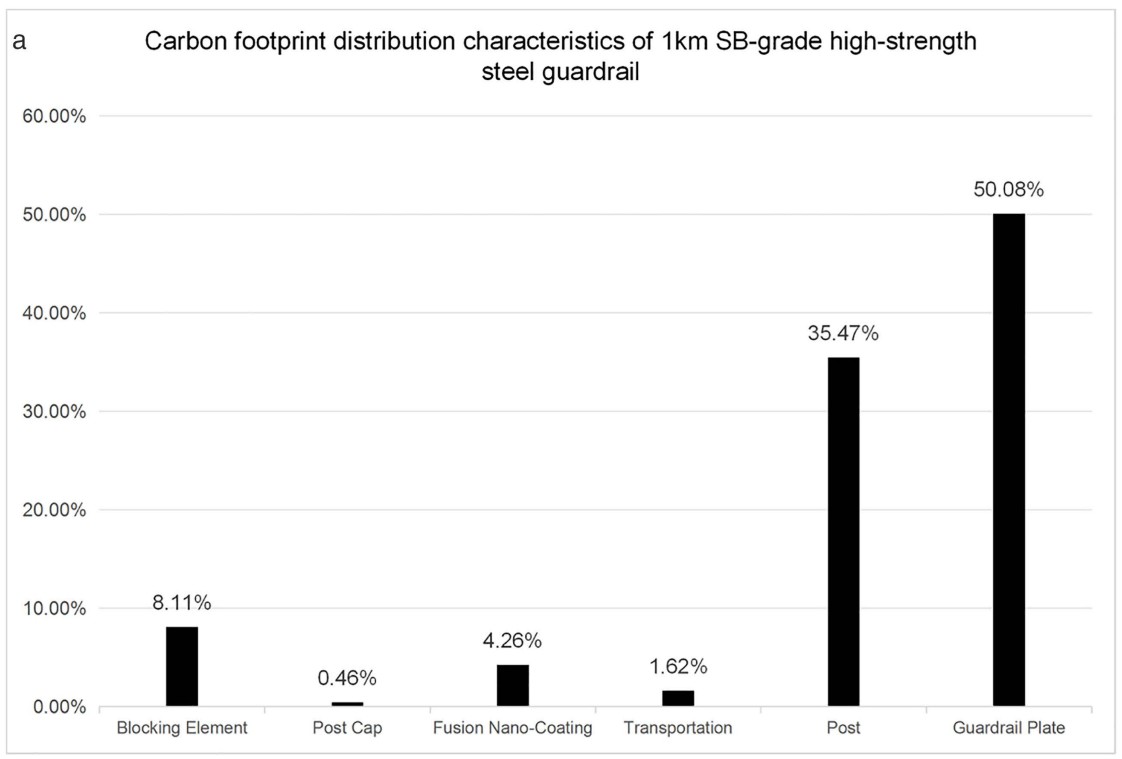

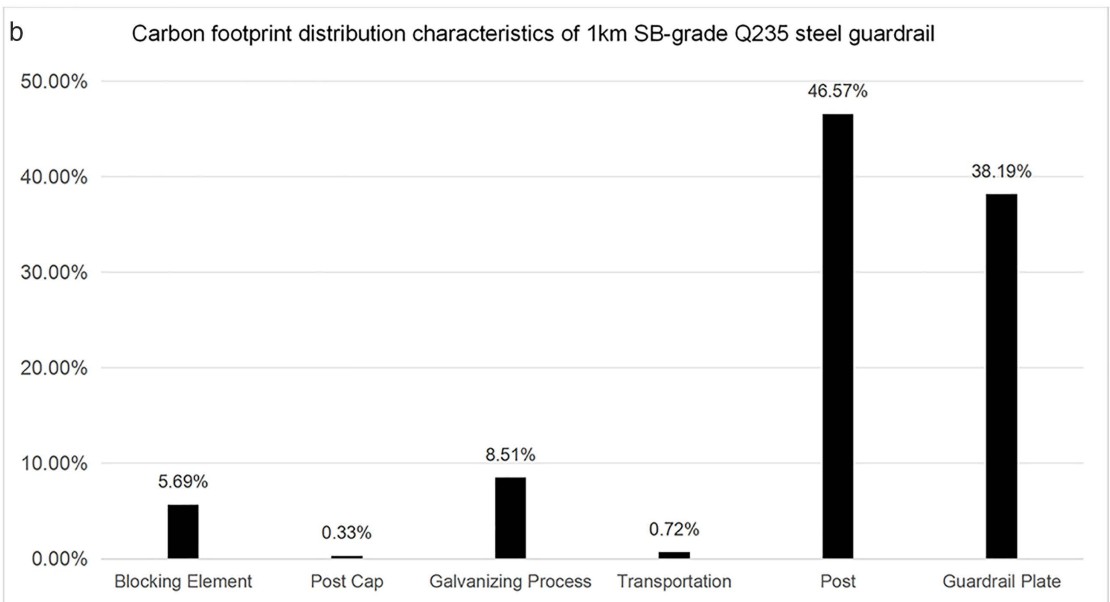

**Fig 1. Carbon Footprint Proportion of Each Stage in the Full Life Cycle of 1 km SB-Grade Steel Guardrails.** (a) Alloy steel corrugated beam guardrail carbon footprint distribution; (b) Q235 steel corrugated beam guardrail carbon footprint distribution.

Buffering/Energy-Absorption Function: For the small passenger car, the post-impact occupant accelerations ($a_x$ = 73.5 m/s², $a_y$ = 126.0 m/s²) and velocities ($V_x$ = 4.3 m/s, $V_y$ = 4.5 m/s) for the alloy steel guardrail were significantly lower than the safety limits and outperformed those of the Q235 steel guardrail. For the medium and large vehicles, the maximum

**Table 7. Sensitivity Analysis Results of Key LCA Parameters.**

| Sensitive Parameter | Baseline Value | Fluctuation | Change in Carbon Footprint (tCO$_2$ e/km) | Change Rate of Carbon Footprint | Sensitivity Ranking | Remarks |
|---|---|---|---|---|---|---|
| Hot Metal Input | 0.9545 t/t-steel | +10% | +6.93 | +8.7% | 1 | Core emission source in the steelmaking stage |
| | | −10% | −6.89 | −8.5% | | |
| Electricity Consumption | Avg. 386.0 kWh/t-steel | +10% | +4.29 | +5.3% | 2 | Comprehensive energy consumption |
| | | −10% | −4.17 | −5.2% | | |
| Natural Gas Consumption | Avg. 1.2 m³/t-steel | +10% | +2.51 | +3.1% | 3 | Primary process fuel |
| | | −10% | −2.48 | −3.1% | | |
| Fusion-Bonded Nano-Powder Usage | 606.99 kg/km | +10% | +1.57 | +1.9% | 4 | Surface treatment material for alloy steel guardrail |
| | | −10% | −1.54 | −1.9% | | |
| Zinc Coating Usage for Galvanizing | 1785.25 kg/km | +10% | +5.82 | +1.9% | 5 | Control group: Q235 steel guardrail |
| | | −10% | −5.77 | −1.9% | | |

Note: The sensitivity ranking is based on the absolute value of the change rate in the carbon footprint. * The analysis for the "Zinc coating usage for galvanizing" parameter applies only to the traditional Q235 steel guardrail in the control group, which uses the hot-dip galvanizing process.

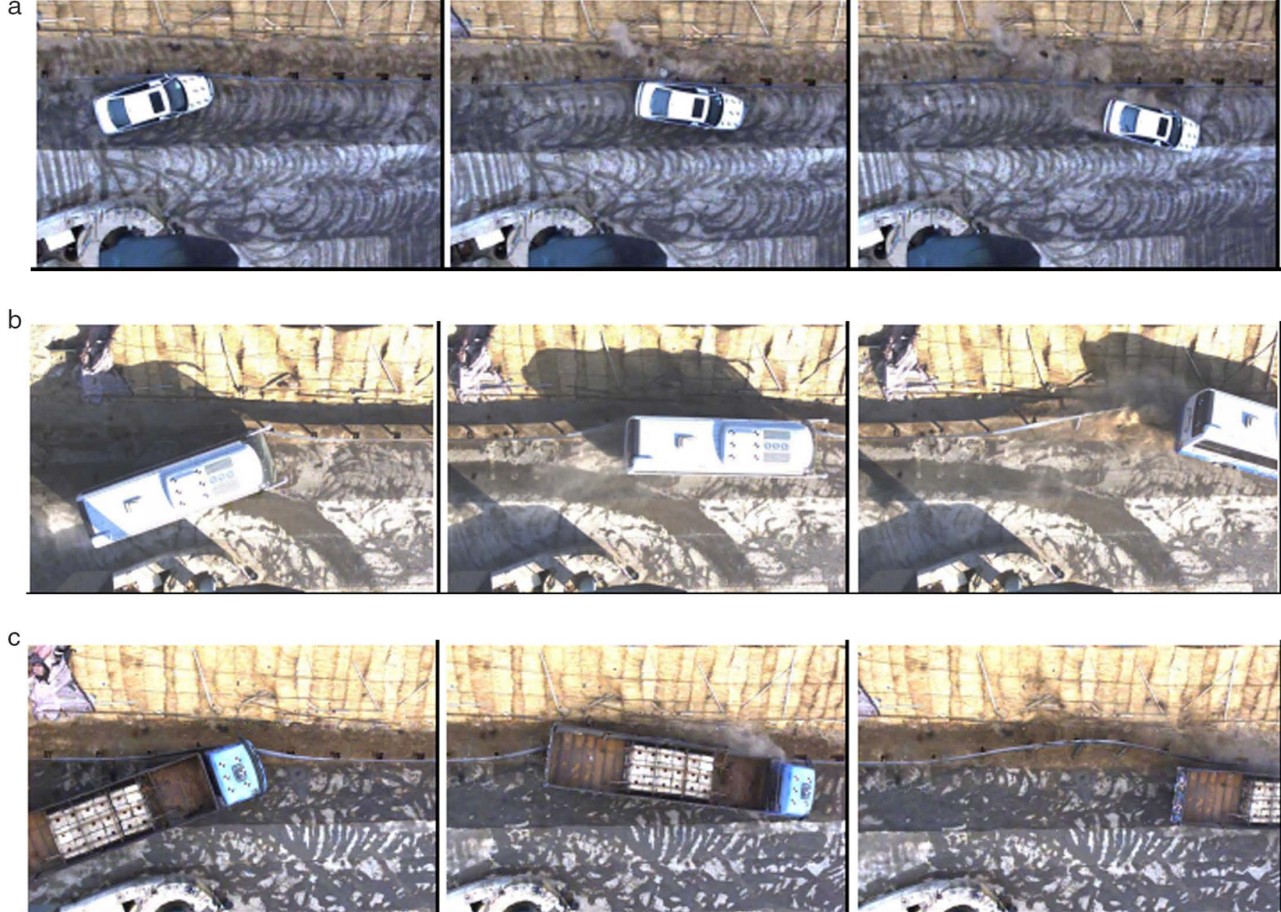

**Fig 2. Process Diagram of Guardrail Impact Test.** (a) Small passenger car; (b) Medium bus; (c) Large truck.

**Table 8. Test Results of Alloy Steel Guardrails vs. Standard Safety Limits.**

| Item | Evaluation Indicator | Safety Limit | Test Group (Alloy Steel) Results | | | Control Group (Q235 Steel) Results | | |
|---|---|---|---|---|---|---|---|---|
| | | | Small Car | Medium Bus | Large Truck | Small Car | Medium Bus | Large Truck |
| Contain-ment | Vehicle must not penetrate, override, or climb over the test guardrail | Not Allowed | Conform | Conform | Conform | Conform | Conform | Conform |
| | Guardrail components and their detached fragments must not intrude into the vehicle occupant compartment | No Intrusion | Conform | Conform | Conform | Conform | Conform | Conform |
| Redirective | Vehicle must not roll over after impact | No Rollover | Conform | Conform | Conform | Conform | Conform | Conform |
| | Vehicle wheel trajectory after impact shall satisfy the guided trajectory envelope requirement | Satisfy Envelope | Conform | Conform | Conform | Conform | Conform | Conform |
| Buffering | Occupant impact longitudinal velocity $V_x$ (m/s) | ≤12 | 4.3 | — | — | 4.6 | — | — |
| | Occupant impact lateral velocity $V_y$ (m/s) | ≤12 | 4.5 | — | — | 5.9 | — | — |
| | Occupant impact longitudinal acceleration $a_x$ (S1, S3 Documents) (m/s²) | ≤200 | 73.5 | — | — | 103.6 | — | — |
| | Occupant impact lateral acceleration $a_y$ (S2, S4 Documents) (m/s²) | ≤200 | 126.0 | — | — | 175.8 | — | — |
| Guardrail max. lateral dynamic deformation (m) | | -- | 1.00 | 1.60 | 1.80 | 0.8 | 1.40 | 1.15 |
| Vehicle max. dynamic outward inclination equivalent value (m) | | -- | — | 1.75 | 2.45 | — | 1.75 | 2.10 |

Note: A dash ("—") indicates that the parameter was not measured or is not applicable for that specific test condition.

dynamic outward inclination equivalent values for the alloy steel guardrail were 1.75 m and 2.45 m, respectively, complying with the standard requirements.

The results indicate that the alloy steel guardrail, despite its lightweight design with 25% and 50% reductions in beam and post thickness, not only fully satisfies the SB-level safety and containment requirements but also exhibits superior performance in certain buffering metrics. This excellent energy absorption capacity is directly attributed to the material's high product of strength and elongation (PSE ≥ 0.175 GPa·%).

## 4. Discussion

### 4.1. Carbon reduction mechanism

The total carbon emission of 1 km of SB-grade alloy steel corrugated beam guardrail is only 80.86 tCO$_2$e, which is about 73.8% (228.28 tCO$_2$e) less than that of the same grade Q235 steel guardrail (309.14 tCO$_2$e). This significant reduction mainly stems from the low-carbon optimization of the alloy steel's ESP production process and the fusion nano-coating process:

ESP Short-Process Carbon Reduction: The alloy steel in this study adopts the ESP endless continuous casting and rolling process, which greatly shortens the process flow, omitting multiple processes such as ingot casting, heating, and roughing in the traditional long process (blast furnace - basic oxygen furnace). Data show that the carbon emission per ton of steel in the traditional long process is about 1.85 tCO$_2$e/t, while the ESP short-process can reduce the unit carbon emission to 1.0-1.48 tCO$_2$e/t steel. In this study, in the steelmaking-rolling link alone, the ESP process reduced the carbon emission intensity by about 1.2 tCO$_2$e/t compared to the traditional long process. Preliminary estimates indicate that the ESP process contributes approximately 60%-70% to the full life cycle carbon reduction.

(2)Fusion Nano-Coating Process Carbon Reduction: The anti-corrosion treatment of the formed alloy steel corrugated beam guardrail adopts the fusion nano-coating process, which uses low-temperature spraying and curing below 100°C,

while the traditional process is hot-dip galvanizing, which requires immersing the steel in molten zinc liquid above 470°C. Compared with hot-dip galvanizing, the comprehensive carbon emission of the fusion nano-coating process is reduced by about 50%-70%. Calculated per ton of guardrail, the carbon emission of traditional hot-dip galvanizing is about 0.5-0.7 $tCO_2e$, while the fusion nano-coating can be reduced to 0.15-0.35 $tCO_2e$. Preliminary estimates indicate that this process replacement contributes approximately 20%-30% to the full life cycle carbon reduction.

In summary, the ESP process innovation and the substitution of low-temperature coating technology together constitute the core of the "dual carbon reduction mechanism" of this solution, and the two synergistically contribute the vast majority of the carbon reduction benefits.

## 4.2. Safety performance improvement

The safety performance of the alloy steel corrugated beam guardrail was validated through full-scale vehicle crash tests. While meeting the SB-level protection standard, the alloy steel guardrail achieved significant lightweighting of the corrugated beam and posts (with thickness reductions of 25% and 50%, respectively). Throughout the impact process, all key occupant risk indicators were superior to the Q235 steel guardrail, and the vehicle's dynamic behavior complied with regulatory requirements. The consistent and reliable safety performance is primarily attributed to the significant enhancement in the guardrail material's properties. The alloy steel possesses a tensile strength exceeding 760 MPa, a lower yield strength over 680 MPa, and maintains an elongation after fracture above 20%. Notably, its tensile strength is approximately 1.5 times that of the traditional Q235 plain carbon steel. The product of strength and elongation (PSE = tensile strength × elongation after fracture) is a key indicator for evaluating the comprehensive mechanical properties and energy absorption capacity of metallic materials. A higher PSE value indicates a greater ability to combine high strength for resisting deformation with good plasticity for absorbing energy under impact. In this study, the PSE of the alloy steel exceeds 0.175 GPa·%, which is higher than the 0.166 GPa·% of the Q235 steel. This elevated PSE provides the material foundation for the guardrail to absorb impact energy through sufficient plastic deformation during a crash. Consequently, even with the lightweight design involving thinner components, the overall protective performance of the guardrail system reliably meets high-standard safety requirements.

## 4.3. Synergistic mechanism of safety and carbon reduction

This study reveals a positive synergistic effect: by adopting high-strength alloy steel and low-carbon processes such as ESP and low-temperature coating, it is not at the cost of sacrificing safety for environmental protection, but rather achieves common improvement of both. The intrinsic mechanism lies in: the high strength and high PSE of the material lay the physical foundation for the improvement of safety performance, while the high-performance material itself makes the guardrail more durable within its life cycle, indirectly reducing carbon emissions caused by replacement and maintenance. At the same time, the advanced short-process and low-temperature processes directly reduce the carbon footprint from the manufacturing end. This indicates that infrastructure upgrades oriented towards the "Dual Carbon" goals can embark on a development path where safety and greenness go hand in hand through front-end material and process innovation.

## 4.4. Cost-benefit analysis

This study conducted a comparative analysis of the per-kilometer initial cost and life-cycle cost (LCC) between alloy steel guardrails and traditional Q235 steel guardrails, both designed to meet the same SB-level safety standard.

Leveraging its high-strength advantage, the alloy steel guardrail enables a lightweight design while achieving the equivalent SB-level safety performance. This design reduces steel consumption by approximately 30% to 34% compared to the Q235 guardrail. This characteristic directly lowers raw material consumption and manufacturing costs. With reference to actual project data, its initial cost per kilometer can be reduced by about 25% compared to the traditional guardrail.

 

From a more comprehensive life-cycle cost (LCC) perspective, the advantages of the alloy steel guardrail are even more pronounced: (1) A substantially extended service life, conservatively estimated in this study to increase from 15 to 25 years; (2) A significant reduction in maintenance frequency and cost, with annual maintenance expenses decreasing from 8,000 CNY/km to 3,000 CNY/km; (3) Notable carbon reduction benefits, achieving a life-cycle carbon reduction of 570.7 $tCO_2e$ per kilometer. Valued at a carbon price of 80 $CNY/tCO_2e$, this translates to a carbon asset value of approximately 45,700 CNY.

The comprehensive calculation demonstrates favorable economic performance for the alloy steel guardrail. Its life-cycle cost per kilometer is 261,000 CNY, which is 13.8% lower than that of the Q235 steel guardrail (303,000 CNY).

## 5. Conclusions

This study leads to the following conclusions on the comprehensive performance of the alloy steel corrugated beam guardrail through the construction of a systematic research framework combining carbon footprint quantification model and safety performance empirical evaluation:

(1) Employing a "low-carbon, high-manganese plus microalloying" compositional design and the ESP short-process route, a 700L-grade alloy steel was successfully produced, exhibiting a tensile strength of 766–781 MPa and a product of strength and elongation (PSE) exceeding 0.175 GPa·%. Its exceptional strength-ductility synergy provides the material foundation for the guardrail to absorb impact energy through substantial plastic deformation. Consequently, while meeting the SB-level safety standard, significant lightweighting of the corrugated beam and posts was achieved (with thickness reductions of 25% and 50%, respectively). This lightweight design, along with its equivalent or superior safety performance, was validated through full-scale vehicle crash tests.

(2) Quantification via an LCA model constructed according to ISO 14067 standards reveals that the carbon footprint of the alloy steel guardrail is 80.86 t $CO_2e$ per kilometer, representing a 73.8% reduction compared to the traditional Q235 steel solution (309.14 t $CO_2e$/km). This breakthrough in carbon reduction is primarily attributed to two key process innovations: First, the ESP (Endless Strip Production) short process, which omits multiple conventional processing steps, reduces the carbon emission intensity per ton of steel by approximately 1.2 t $CO_2e$/t and contributes about 65% of the total reduction. Second, the substitution of energy-intensive hot-dip galvanizing with a low-temperature fusion-bonded nano-coating process, which accounts for roughly 25% of the reduction. Sensitivity analysis further identifies hot metal input and electricity consumption as the most critical parameters for further optimizing the carbon footprint in the future.

(3) A comprehensive cost model incorporating initial construction, operation, maintenance, and environmental benefits demonstrates that the alloy steel guardrail achieves a life-cycle cost of 261,000 CNY per kilometer, which is 13.8% lower than that of the traditional guardrail (303,000 CNY/km). This economic advantage stems from material savings enabled by lightweighting (a 30%–34% reduction in steel tonnage), a substantially extended service life (increased from 15 to 25 years), and lower maintenance requirements. This proves that, within the context of the "Dual Carbon" goals, the moderate upfront investment in high-performance materials and low-carbon processes can be offset by long-term operational benefits in terms of carbon, energy, and material savings, underscoring its significant potential for techno-economic promotion.

## Supporting information

**S1 File. SB-grade alloy steel guardrail acceleration record (X axis).**
(TXT)

**S2 File. SB-grade alloy steel guardrail acceleration record (Y axis).**
(TXT)

**S3 File. SB-grade Q235 steel guardrail acceleration record (X axis).**
(TXT)

**S4 File. SB-grade Q235 steel guardrail acceleration record (Y axis).**
(TXT)

## Acknowledgments

The authors gratefully acknowledge the support from the research team and the enterprises involved in providing the production data.

## Author contributions

**Conceptualization:** Shuai Gong.

**Data curation:** Shuai Gong, Wendong Fan, Haoze Zhao.

**Formal analysis:** Shuai Gong.

**Investigation:** Wendong Fan, Haoze Zhao.

**Methodology:** Zhihao Zhang.

**Project administration:** Wendong Fan, Zhihao Zhang, Shuming Yan.

**Resources:** Zhihao Zhang, Shuming Yan.

**Software:** Shuming Yan.

**Validation:** Shuai Gong, Wendong Fan.

**Visualization:** Shuai Gong, Zhihao Zhang.

**Writing – original draft:** Shuai Gong.

**Writing – review & editing:** Shuai Gong, Shuming Yan.

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
