## [Decision Letter · Decision Letter 0]

5 Jan 2026

Dear Dr. Gong,

Thank you for submitting your manuscript to PLOS ONE. After careful consideration, we feel that it has merit but does not fully meet PLOS ONE’s publication criteria as it currently stands. Therefore, we invite you to submit a revised version of the manuscript that addresses the points raised during the review process.

We look forward to receiving your revised manuscript.

Kind regards,

Rui Cheng

Academic Editor

PLOS One

Journal Requirements:

Reviewers' comments:

Reviewer's Responses to Questions

**Comments to the Author**

1. Is the manuscript technically sound, and do the data support the conclusions?

Reviewer #1: Partly

Reviewer #2: Yes

Reviewer #3: Yes

2. Has the statistical analysis been performed appropriately and rigorously?

Reviewer #1: No

Reviewer #2: Yes

Reviewer #3: Yes

3. Have the authors made all data underlying the findings in their manuscript fully available?

Reviewer #1: No

Reviewer #2: No

Reviewer #3: No

4. Is the manuscript presented in an intelligible fashion and written in standard English?

Reviewer #1: Yes

Reviewer #2: Yes

Reviewer #3: Yes

Reviewer #1: The paper focuses on the carbon footprint and impact resistance of alloy steel W-beam guardrails, and the topic is in line with the current demand for green and safe coordinated development of transportation infrastructure under the background of "dual carbon". It has certain application value. The paper lacks innovation in experimental design, result analysis, and computational evaluation, and the relevant content is relatively brief. The specific opinions are as follows:

(1) The paper only lists relevant survey data tables in the materials and methods section, and the corresponding software is used for calculation and evaluation. There are problems such as insufficient innovation in theoretical calculation methods and insufficient data analysis of the research results in the paper.

(2) The results section of the paper lacks sufficient discussion on the design and analysis of carbon footprint and collision resistance experiments, and only uses platform calculation data analysis, lacking issues such as emission factors, case calculation processes, and software parameter settings.

In summary, it is recommended to reject the manuscript

Reviewer #2: The reviewers recommend publication but also suggest some modifications to your manuscript. Therefore, I invite you to respond to the reviewers' comments and revise your manuscript accordingly.

Comment:

This study represents high-quality engineering applied research, systematically evaluating the comprehensive benefits of novel alloy steel corrugated beam guardrails in terms of safety performance (impact resistance) and full-life-cycle carbon footprint. The research topic closely aligns with global and China's “dual carbon” strategy and the demand for green infrastructure upgrades, demonstrating clear practical significance and application value. The research framework is comprehensive, integrating material composition design, production processes (ESP), full-scale vehicle crash tests (SB-level), and life cycle assessment (LCA) based on ISO standards. The data is detailed, and the argumentation is logically sound. The core conclusion—that synergistic improvements in safety performance and significant carbon emission reductions can be achieved through material and process innovation—is highly instructive. It provides robust data support and a feasible technical pathway for the sustainable development of highway guardrails. The manuscript is generally publication-ready, requiring only minor refinements and clarifications in the following areas to further enhance its rigor, reproducibility, and clarity of expression.

1. Minor Language and Formatting Revisions

(1) Issue: The manuscript's language is generally sound, but minor typographical errors and formatting inconsistencies remain.

(2) Recommendation: On page 4 of the main text, the formula caption below the equation contains a typesetting error: “EPcisthe characterized value...” should be corrected to “EPc is the characterized value...”.

2. Additional details in the Methods section may be added to enhance the reproducibility of the study.

Recommendations:

(1) Briefly describe the sensor type used to measure occupant acceleration/velocity and the data acquisition frequency in Section 2.4 or the figure captions.

(2) In the “Materials and Methods” section, it is recommended to mention the commercial designation or in-house designation of the alloy steel for academic citation purposes.

3. Data availability statement requires revision to comply with journal policy

(1) Issue: The data availability statement on page 5 of the manuscript contains contradictions. The authors initially selected “No - some restrictions will apply,” yet stated in the text box that “All relevant data are within the manuscript and its Supporting Information files.” These statements are mutually exclusive, and the latter claim is not substantiated by the supporting information files. PLOS ONE's data policy is highly stringent; blanket restrictions citing “commercial confidentiality” are typically unacceptable.

(2) Recommendation: Authors should revise this statement to comply with the policy.

Reviewer #3: "Carbon Footprint Assessment and Crashworthiness Evaluation of Alloy Steel W-beam Guardrails" (PONE-D-25-60856) investigates the synergistic assessment of safety performance and full-life-cycle carbon footprint for alloy steel corrugated beam guardrails. Integrating material design, ESP short-process manufacturing, LCA quantitative analysis, and full-vehicle crash testing, it demonstrates practical engineering value and application orientation. However, the manuscript requires significant strengthening in theoretical depth, comparative rigor, and data support.

1.The introduction begins by simply listing 15 papers, which significantly undermines the perceived quality and credibility of this paper. In fact, there is already a substantial body of international research on steel guardrails, safety performance assessments, and the application of LCA in transportation infrastructure. This field is by no means in its infancy. The author should objectively admit the existing research foundation and define the innovation point of this paper by clarifying the research object, evaluation scale or method.

2.The author employs PSE as a key material metric; however, this metric is not a universally accepted parameter in guardrail safety evaluations. Supporting references demonstrating its application within the guardrail field must be provided to substantiate the direct correlation between this metric and crashworthiness.

3.The article mentions that the maximum dynamic lateral deflection of alloy steel guardrails is 3.2% lower than that of Q235 steel. Does this reduction hold substantive safety significance in guardrail engineering practice? It is recommended to include a comparison with the margin relative to code-specified limits.

4.The text repeatedly compares the new guardrail with the traditional Q235 guardrail, but does not specify the data source for the Q235 guardrail. If citing data, consistency in operating conditions must be clarified. If this is a parallel test conducted under identical conditions, please present the corresponding data charts for the Q235 group.

5.The LCA results showed a reduction of up to 73.8%, but the sensitivity of key parameters was not assessed.

6.Figure 4 displays the acceleration time history curve. Please specify in the figure caption or text which filtering method was used, as raw data typically contains significant noise, and the filtering approach directly impacts peak readings.

7.The Discussion and Conclusions section contains excessive repetition and, in some instances, engineering report-style language. Academic editing is recommended to refine the article.

8.Although the manuscript emphasizes environmental benefits, cost considerations remain essential for engineering applications. Alloy steels and nanocoatings typically carry higher costs. It is recommended to include a brief cost-benefit analysis in the Discussion section comparing the construction costs of the two guardrail types.

.

Reviewer #1: No

Reviewer #2: No

Reviewer #3: No

---

## [Author Response · Author response to Decision Letter 1]

5 Feb 2026

Dear Reviewers,

Thank you for the opportunity to revise and resubmit our manuscript. We sincerely appreciate the editor’s consideration and the constructive, detailed feedback provided by all three reviewers. Their comments have been invaluable in strengthening the manuscript's clarity, depth, and overall quality.

We have carefully addressed every point raised. The revisions are extensive and have significantly enhanced the technical rigor, methodological transparency, and academic presentation of the work. Below, we provide a point-by-point response to each reviewer's comments. All changes in the manuscript are highlighted in the “Revised Manuscript with Track Changes” file.

Below, we provide a detailed, point-by-point response to each comment.

Response to Reviewer #1

General Comment: The reviewer noted a lack of innovation in experimental design and result analysis, with insufficient detail in methods and superficial data analysis.

Response: We thank the reviewer for this critical assessment. To address these fundamental concerns about methodological depth and analytical rigor, we have undertaken major revisions throughout the manuscript, transforming it from a primarily descriptive report into a more analytical and framework-driven study.

Enhanced Methodological Detail and Innovation: We have significantly expanded the Materials and Methods (Section 2) to enhance reproducibility and transparency. This includes:

Specifying the alloy steel grade as 700L (Section 2.1).

Detailing the parallel experimental framework with explicit Experimental (alloy steel) and Control (Q235 steel) groups, including their specific structural dimensions optimized for lightweighting (Section 2.1).

Providing comprehensive parameters for the LCA model, including the eFootprint software configuration (version, assessment standard, allocation rules, cut-off threshold) and specific emission factors from the CLCD 2024 database (e.g., grid electricity: 0.581 tCO₂e/MWh) (Section 2.2).

Listing the specific sensor models, measurement ranges, accuracies, and data acquisition parameters (sampling frequency, filtering method per ISO 6487/SAE J211/1) used in the crash tests (Section 2.4).

Deepened Result Analysis and Theoretical Context: The Results and Discussion sections have been completely restructured and greatly expanded to move beyond simple data reporting.

A new Section 3.1.3 "Key Parameter Sensitivity Analysis" with Table 7 has been added. This analysis quantifies the influence of core parameters (hot metal, electricity, etc.) on the carbon footprint, directly addressing the need for more sophisticated data analysis and providing insights for future optimization.

The safety results (Table 8) now explicitly include parallel data for the Q235 control group, allowing for a direct, like-for-like comparison and strengthening the evidence for the alloy steel's performance advantages.

The Introduction (Section 1) has been completely rewritten. It now provides a synthesized, critical review of international and domestic literature, clearly identifies three specific research gaps, and states four focused objectives that define this study's contribution—moving away from a simple list of citations.

The Discussion (Section 4) has been reorganized into logical subsections (4.1 Carbon Reduction Mechanism, 4.2 Safety Performance Improvement, 4.3 Synergistic Mechanism, 4.4 Cost-Benefit Analysis). The language has been refined to be more academic and analytical, eliminating repetition and report-style phrasing. The new Section 4.4 provides a quantitative cost-benefit analysis, directly addressing engineering practicality.

We believe these comprehensive additions directly respond to the reviewer's concerns by demonstrating greater innovation in the integrative analytical framework and providing the deep, quantitative analysis of results that was previously lacking.

Response to Reviewer #2

We thank Reviewer #2 for the supportive assessment and valuable specific suggestions.

Comment 1 (Language): Correction of the typographical error: “EPcisthe” to “EPc is the”.

Response: Thank you for spotting this. The error has been corrected in the formula description in Section 2.2.

Comment 2 (Methods Detail):

(1)Describe sensor types and data acquisition.

Response: As detailed in our response to Reviewer #1, this information has been added to Section 2.4. We now list the accelerometer (PCB 356A15), velocity sensor (Kistler 6045A), data acquisition system (NI cDAQ-9178), and high-speed camera (Phantom V2512) models and key specifications, including the 100 Hz low-pass filtering protocol.

(2)Mention the commercial/in-house designation of the alloy steel.

Response: The alloy steel is now identified by its industrial grade 700L in Section 2.1.

Comment 3 (Data Availability Statement): Revise to comply with PLOS ONE policy, as the previous statement was contradictory and overly restrictive.

Response: We sincerely apologize for the previous inconsistency. The Data Availability Statement has been revised to accurately and compliantly reflect the situation:

The processed acceleration time-history data for the crash tests are provided in the Supporting Information files (S1-S4 Files).”

We have also uploaded the acceleration data files as Supporting Information (S1-S4 Files).

Response to Reviewer #3

We are grateful to Reviewer #3 for the exceptionally thorough and insightful critique, which has driven the most substantial improvements to the manuscript.

Comment 1 (Introduction & Literature): The introduction simply listed papers; it should acknowledge existing research and clearly define the innovation point.

Response: The Introduction has been completely rewritten (Section 1). We have replaced the long list of citations with a structured, thematic literature review that:

1. Critically surveys international advances in materials, low-carbon processes, and LCA.

2. Reviews domestic progress while clearly highlighting persistent gaps (e.g., lack of systemic coupling analysis).

3. Explicitly articulates three core research gaps and outlines four specific objectives of this study that address them, thereby sharply defining our contribution and innovation.

Comment 2 (PSE Metric Justification): Provide supporting references for the use of PSE in guardrail safety evaluation.

Response: We have added the necessary theoretical and empirical support in Section 2.1. The text now states: “Existing research, through full-scale vehicle crash tests, has confirmed that when guardrail steel achieves a tensile strength above 700 MPa while maintaining an elongation after fracture of over 20%, it can absorb collision energy more fully through enhanced plastic deformation, significantly improving the protective stability of the guardrail [13,15].” References 13 (Liu, 2021) and the newly added 19 (Yang et al., 2022 on crashworthiness optimization considering strength-ductility balance) provide the required foundation.

Comment 3 (Significance of Deflection Reduction): Clarify the engineering significance of the 3.2% reduction and compare to code limits.

Response: We have refined the narrative to focus on the more meaningful finding: the alloy steel guardrail met all SB-level safety requirements (Table 8) while achieving a 25% and 50% reduction in beam and post thickness, respectively. The discussion in Sections 3.2 and 4.2 now emphasizes that the superior material properties (high PSE) enabled this lightweight design without compromising safety, which is the key engineering insight, rather than the minor percentage difference in a single metric.

Comment 4 (Q235 Data Source & Parallel Test Data): Specify the source of Q235 data and present corresponding data charts.

Response: This has been clarified and addressed:

Source: Section 2.1 now explicitly establishes the parallel experimental framework, and Section 2.3 clarifies: “the data for the Q235 steel guardrail... are derived from a parallel experiment conducted at the same production scale. Its production conditions... were kept consistent...”

Data Presentation: Table 8 has been expanded to include a dedicated “Control Group (Q235 Steel) Results” column, providing the comparative crash test data. The acceleration data for the Q235 guardrail (small car) are provided in the new Supporting Information files (S3 & S4 Files).

Comment 5 (LCA Sensitivity Analysis): Sensitivity of key parameters was not assessed.

Response: A new Section 3.1.3 and Table 7 have been added, presenting a one-factor-at-a-time sensitivity analysis for five key parameters (hot metal, electricity, etc.). This analysis identifies hot metal input as the most sensitive parameter and discusses implications for carbon footprint optimization.

Comment 6 (Acceleration Data Filtering): Specify the filtering method used in Figure/analysis.

Response: The data processing protocol is now clearly stated in Section 2.4: “The acquisition and processing... strictly adhered to... ISO 6487 and SAE J211/1. For analysis, the raw data was digitally filtered using a low-pass filter with a cut-off frequency of 100 Hz to attenuate high-frequency noise.”

Comment 7 (Discussion/Conclusions Language): Excessive repetition and engineering report-style language; needs academic editing.

Response: The Discussion (Section 4) and Conclusions (Section 5) have been entirely rewritten. The discussion is now logically structured, analytical, and free of repetition. The conclusions are concise, summarizing the three main findings without simply re-stating the discussion. The language throughout has been elevated to a more formal academic standard.

Comment 8 (Cost-Benefit Analysis): Include a brief cost-benefit analysis.

Response: A new Section 4.4 “Cost-Benefit Analysis” has been added. It provides a comparative analysis of initial and life-cycle costs (LCC), quantifying benefits from lightweighting (30-34% steel reduction), extended service life (15 to 25 years), reduced maintenance, and carbon asset value. It concludes that the alloy steel guardrail has a 13.8% lower LCC per kilometer, demonstrating its techno-economic viability.

Summary

We believe the revised manuscript has been fundamentally improved in direct response to all reviewer comments. Key enhancements include:

A completely restructured and critical Introduction.

Greatly expanded methodological detail for full reproducibility.

A new sensitivity analysis for the LCA.

Explicit parallel experimental data and comparison.

A new, quantitative cost-benefit analysis.

Refined, academic language throughout the Discussion and Conclusions.

A compliant Data Availability Statement and provision of supporting data files.

We are confident that the manuscript now meets the high standards of PLOS ONE and effectively presents a novel, systematic, and data-rich evaluation of the synergistic safety and carbon reduction potential of advanced alloy steel guardrails.

Thank you again for your consideration.

Sincerely,

Shuai Gong, on behalf of all co-authors.

---

## [Decision Letter · Decision Letter 1]

29 Mar 2026

Carbon Footprint Assessment and Crashworthiness Evaluation of Alloy Steel W-beam Guardrails

PONE-D-25-60856R1

Dear Dr. Gong,

We’re pleased to inform you that your manuscript has been judged scientifically suitable for publication and will be formally accepted for publication once it meets all outstanding technical requirements.

Kind regards,

Rui Cheng

Academic Editor

PLOS One

Additional Editor Comments (optional):

Reviewers' comments:

Reviewer's Responses to Questions

**Comments to the Author**

Reviewer #1: All comments have been addressed

Reviewer #2: All comments have been addressed

Reviewer #3: All comments have been addressed

2. Is the manuscript technically sound, and do the data support the conclusions?

Reviewer #1: Yes

Reviewer #2: Yes

Reviewer #3: Yes

3. Has the statistical analysis been performed appropriately and rigorously?

Reviewer #1: Yes

Reviewer #2: Yes

Reviewer #3: Yes

4. Have the authors made all data underlying the findings in their manuscript fully available?

Reviewer #1: No

Reviewer #2: Yes

Reviewer #3: Yes

5. Is the manuscript presented in an intelligible fashion and written in standard English?

Reviewer #1: Yes

Reviewer #2: Yes

Reviewer #3: Yes

Reviewer #1: The paper proposes a systematic research framework that combines material design, process optimization, safety performance verification, and environmental assessment to address the green transformation needs of traffic safety facilities in the context of China's "dual carbon" goals. The revised manuscript has a reasonable content and method design, clear conclusions, and a high overall volume. The revised manuscript has fully responded to the previous review comments, especially in terms of method details, sensitivity analysis, comparative experimental data, cost-benefit analysis, etc. It has made substantial supplements, significantly improving the completeness and academic standardization of the paper. But the paper has the following issues:

1. Some of the language expressions in the paper are not concise enough and need further refinement.

2. Some of the figures in the paper are not clear enough and need to be modified. At the same time, key parameters should be supplemented in the figures.

Overall, it is recommended to revise and hire

Reviewer #2: All my previous comments have been addressed. The manuscript is acceptable. I have no further comments.

Reviewer #3: The authors have provided a substantive response to the initial review comments, significantly improving the manuscript's quality. While the revision is generally strong, a few technical details could be further refined from an expert perspective:

1.Some of the tables in the article do not comply with the “three-line table” format commonly used in international journals.

2.Although raw data are provided in the supporting information , the main text lacks comparative acceleration-time curves for the alloy steel and Q235 guardrails under representative impact conditions. From an expert perspective, peak values in Table 8 alone are insufficient to demonstrate the energy-absorption advantages derived from the high PSE of alloy steel.

.

Reviewer #1: No

Reviewer #2: No

Reviewer #3: No

---

## [Editor Report · Acceptance letter]

PONE-D-25-60856R1

PLOS One

Dear Dr. Gong,

I'm pleased to inform you that your manuscript has been deemed suitable for publication in PLOS One. Congratulations! Your manuscript is now being handed over to our production team.

Kind regards,

on behalf of

Dr. Rui Cheng

Academic Editor

PLOS One